# Wound Infections of Snakebites from the Venomous *Protobothrops mucrosquamatus* and *Viridovipera stejnegeri* in Taiwan: Bacteriology, Antibiotic Susceptibility, and Predicting the Need for Antibiotics—A BITE Study

**DOI:** 10.3390/toxins12090575

**Published:** 2020-09-07

**Authors:** Chih-Chuan Lin, Yen-Chia Chen, Zhong Ning Leonard Goh, Chen-Ken Seak, Joanna Chen-Yeen Seak, Gao Shi-Ying, Chen-June Seak

**Affiliations:** 1Department of Emergency Medicine, Lin-Kou Medical Center, Chang Gung Memorial Hospital, Taoyuan 33305, Taiwan; bearuncle@yahoo.com (C.-C.L.); s78092359@cgmh.org.tw (G.S.-Y.); julianseak@sem.org.tw (S.I.); 2College of Medicine, Chang Gung University, Taoyuan 33302, Taiwan; 3Department of Emergency medicine, Taipei Veterans General Hospital, Taipei 11217, Taiwan; ycchen4@vghtpe.gov.tw; 4Department of Emergency Medicine, School of Medicine, National Yang-Ming University, Taipei 11221, Taiwan; 5Sarawak General Hospital, Kuching, Sarawak 93586, Malaysia; lgzn92@gmail.com (Z.N.L.G.); jonathanseak@gmail.com (C.-K.S.); joannaseak@hotmail.com (J.C.-Y.S.); 6Department of Emergency Medicine, New Taipei Municipal Tucheng Hospital, New Taipei City 23652, Taiwan

**Keywords:** wound infections, snakebites, Taiwan habu, *Protobothrops mucrosquamatus*, green bamboo viper, *Viridovipera stejnegeri*, Bacteriology of Infections in Taiwanese snake Envenomation (BITE), Stratification to Prevent Overcrowding Taskforce (SPOT)

## Abstract

Snakebites from Taiwan habus (*Protobothrops mucrosquamatus*) and green bamboo vipers (*Viridovipera stejnegeri*) account for two-thirds of all venomous snakebites in Taiwan. While there has been ongoing optimization of antivenin therapy, the proper management of superimposed bacterial wound infections is not well studied. In this Bacteriology of Infections in Taiwanese snake Envenomation (BITE) study, we investigated the prevalence of wound infection, bacteriology, and corresponding antibiotic usage in patients presenting with snakebites from these two snakes. We further developed a BITE score to evaluate the probability of wound infections and guide antibiotic usage in this patient population. All snakebite victims who presented to the emergency departments of seven training and research hospitals and received at least one vial of freeze-dried hemorrhagic antivenin between January 2001 and January 2017 were identified. Patient biodata, laboratory investigation results, and treatment modalities were retrieved. We developed our BITE score via univariate and multiple logistic regression analyses. The receiver operating characteristic (ROC) curve was plotted to evaluate the predictive performance of the BITE score. Out of 8,295,497 emergency department visits, 726 patients presented with snakebites from a Taiwan habu or a green bamboo viper. The wound infection rate was 22.45%, with seven positive wound cultures, including six polymicrobial infections. *Morganella morganii*, *Enterococcus* spp., *Bacteroides fragilis*, and *Aeromonas hydrophila* were most frequently cultured. There were no positive blood cultures. A total of 33.0% (*n* = 106) of snakebite patients who received prophylactic antibiotics nevertheless developed wound infections, while 44.8% (*n* = 73) of wound infection patients were satisfactorily treated with one of the following antibiotics: amoxicillin/clavulanic acid, oxacillin, cefazolin, and ampicillin/sulbactam. With the addition of gentamicin, the success of antibiotic therapy increased by up to 66.54%. The prognostic factors for the secondary bacterial infection of snakebites were white blood cell counts, the neutrophil lymphocyte ratio, and the need for hospital admission. The area under the ROC curve for the BITE score was 0.839. At the optimal cut-off point of 5, the BITE score had a 79.58% accuracy, 82.31% sensitivity, and 79.71% specificity when predicting infection in snakebite patients. Our BITE score may help with antibiotic stewardship by guiding appropriate antibiotic use in patients presenting with snakebites. It may also be employed in further studies into antibiotic prophylaxis in snakebite patients for the prevention of superimposed bacterial wound infections.

## 1. Introduction

Snakebites from Taiwan habus (*Protobothrops mucrosquamatus*, previously known as *Trimeresurus mucrosquamatus*) and green bamboo vipers (*Viridovipera stejnegeri*, previously known as *Trimeresurus gramineus*) account for two-thirds of all venomous snakebites in Taiwan [1,2,3]. The management of envenomation by these two snakes is similar due to various similarities in their clinical presentation. Patients often present to the emergency department (ED) with symptoms and signs mimicking cellulitis and infection, such as localized tissue swelling, tenderness, and warmth with elevated white blood cell (WBC) counts [4]. Since clinical differentiation between envenomation and wound infection is difficult, especially in early presentations, such snakebites are often routinely treated with prophylactic antibiotics [5], in addition to the freeze-dried hemorrhagic (FH) antivenin [6,7]. This practice of administering prophylactic antibiotics is widespread, in line with the general management of animal bites to prevent the development of subsequent wound infections.

Up to 80% of these snakebite patients are not eventually diagnosed with cellulitis or a wound infection [5], representing a gross overuse of antibiotics in this patient population. This issue arises from the current lack of evidence-based guidelines or tools to guide proper antibiotic therapy in snakebite patients. As such, developing a suitable tool to predict the probability of developing a snakebite wound infection is crucial for augmenting the management of snakebite complications. These prognostic tools may also be employed in further studies to investigate the choice of antibiotic prophylaxis in snakebite patients. We thus embarked on this BITE (Bacteriology of Infections in Taiwanese snake Envenomation) study to investigate the prevalence of wound infection in snakebites from the two aforementioned snakes, the bacteriology of these wound infections, and the corresponding antibiotic usage. The ultimate aim was to develop a prognostic evaluation tool to guide antibiotic usage in patients with snakebites from the Taiwan habu and the green bamboo viper, where the higher the risk of infection, the stronger the indication for antibiotic prophylaxis. This would then provide clinicians with a framework to prescribe antibiotics judiciously.

## 2. Results

### 2.1. Patients’ Characteristics

Of the 8,295,497 ED patient encounters retrieved from the Chang Gung Research Database (CGRD), there were 1128 snakebite patients (0.01% of total ED visits), of whom, 726 received only the FH antivenin (64.36% of total snakebite patients), i.e., 726 patients were bitten by either the Taiwan habu or the green bamboo viper. Males accounted for the majority of this study population (*n* = 506, 69.7%), with a mean age of 52 years. A median of two vials of FH antivenin (IQR 1–3) was administered to each patient, suggesting that these patients were mostly mildly to moderately envenomed. A total of 230 patients (31.68%) required hospital admission with a mean hospital stay of 7.5 days.

A total of 163 of these 726 recruited patients (22.45%) eventually developed wound infections. These patients with infected snakebite wounds had longer hospital stay lengths (Table 1), implying that they were in more severe clinical conditions compared to patients without infected wounds [5]. The patients with the most severe envenomation consequently had the most severely infected wounds, which required surgical debridement.

While 117 patients with infected snakebite wounds were satisfactorily treated after antivenin therapy with or without antibiotics, 46 required surgical intervention. In particular, there were 34 wound debridements, 42 fasciotomies, and 26 grafts done; 15 of these 46 patients required all three surgical interventions during their hospital admission. Patients who underwent wound debridement were older (57.82 ± 15.74 vs. 51.79 ± 17.59, *p* = 0.04) and had higher WBC × neutrophil-lymphocyte ratios (61.85 ± 74.26 vs. 37.69 ± 70.41, *p* = 0.07), while those who underwent fasciotomies had higher WBC × neutrophil-lymphocyte ratios (73.57 ± 73.31 vs. 36.44 ± 69.97, *p* = 0.003). There were no deaths or limb disabilities.

Swab cultures for both aerobic and anaerobic bacteria were performed during the debridement or fasciotomy procedures. They were not routinely obtained for patients clinically diagnosed with cellulitis, unless prominent pus or wound discharge was present, hence suggesting infection. Only 7 patients had positive swab cultures, 6 of which demonstrated polymicrobial infections implicating 12 organisms. Repeated cultures during each surgical procedure produced similar results (Table 2).

All isolated pathogens, except *Stenotrophomonas maltophilia* and *Staphylococcus saprophyticus*, were skin commensals or opportunistic pathogens. Blood cultures were routinely performed for all snakebite patients receiving antibiotic prophylaxis before the antibiotic administration, though none returned positive.

Based on our antibiotic sensitivity results, we have suggested antibiotics of choice for each isolate to be used in the daily clinical practice of our center (Table 2). Full details regarding the antibiotic susceptibility testing for each of the isolated microorganisms are included in our Appendix A.

### 2.2. Bacteriology and the Possible Influence of Prophylactic Antibiotics

Gram-positive bacteria were more frequently identified than Gram-negative and anaerobic bacteria. The three most frequently cultured Gram-positive bacteria were *Enterococcus faecalis*, *Staphylococcus* spp., and *Corynebacterium* spp., while the most common Gram-negative bacteria cultured was *Morganella morganii.* The combination of *Enterococcus faecalis* and *Morganella morganii* was most common in polymicrobial infections (Table 2).

The yield of positive microbiological cultures in our study was much lower than expected, especially when compared with other studies. This could perhaps be attributed to the systematic pre-emptive use of antibiotics in snakebite patients on ED presentation in our setting.

### 2.3. Antibiotic Susceptibility

The results of the antibiotic sensitivity tests have been included in Appendix A. Gram-positive bacteria were susceptible to ampicillin, ciprofloxacin, levofloxacin, amikacin, vancomycin, and teicoplanin. Vancomycin was our drug of choice for treating oxacillin-resistant strains of *Staphylococcus aureus*. Coagulase-negative *Staphylococcus* were sensitive to teicoplanin and vancomycin.

The Gram-negative bacteria isolated in our study were all susceptible to ampicillin, piperacillin/tazobactam, ceftriaxone, ceftazidime, ciprofloxacin, levofloxacin, gentamicin, amikacin, trimethoprim-sulfamethoxazole, vancomycin, and teicoplanin. Our drugs of choice for wound infections with *Corynebacterium* species include fluoroquinolones, carbapenems, teicoplanin, and vancomycin.

Anaerobic bacteria, such as *Peptostreptococcus micros* and *Bacteroides fragilis*, were susceptible to clindamycin and metronidazole. *Enterobacter cloacae* were susceptible to ciprofloxacin, levofloxacin, amikacin, vancomycin, and teicoplanin.

### 2.4. Use of Prophylactic Antibiotics

A total of 321 patients (44.2%) received oral antibiotic prophylaxis. Snakebite patients in the wound infection group (62.72%) were prescribed antibiotics prophylactically more frequently than those without a wound infection (38.6%) (Table 1). The antibiotic prophylaxis chosen was usually cefadroxil (*n* = 100), amoxicillin/clavulanic acid (*n* = 98), or dicloxacillin (*n* = 88).

### 2.5. Antibiotic Therapy for Infected Snakebite Wounds

A total of 73 patients with infected wounds (44.8%) were treated with a single antibiotic, most commonly amoxicillin/clavulanic acid or oxacillin (*n* = 19, 11.95% each). Other single-agent antibiotics used included cefazolin (*n* = 16, 10.06%), ampicillin/sulbactam (*n* = 5, 3.14%), cephradine, ceftriaxone, cefuroxime, clindamycin, and gentamicin.

Of the various poly-agent antibiotic regimens used, the combination of cefazolin and gentamicin was most favored (*n* = 21, 13.2%), with oxacillin and gentamicin being the second most favored (*n* = 12, 7.54%).

### 2.6. Development of the BITE Score

The univariate analyses found patients with infected snakebite wounds to have higher WBC counts (*p* = 0.0008), higher segment counts (*p* = 0.0003), lower lymphocyte counts (*p* = 0.0001), and higher neutrophil-lymphocyte ratios (*p* = 0.01). The 163 patients with wound infections were also found to receive a higher antivenin dose (*p* < 0.0001), have higher admission rates (*p* < 0.0001), and longer lengths of hospitalization (*p* < 0.0001) (Table 1).

These statistically significant variables were then further analyzed via multiple logistic regression to identify independent predictors of wound infections, namely, WBC (in 10^3^/µL) × neutrophil-lymphocyte ratio (NLR) ≥ 19.84 and hospital admission (Table 3).

The β coefficient of each variable was divided by 0.4 and rounded off to the nearest integer to form our BITE score. The stratification of our study population according to their BITE scores demonstrated increasing infection rates (Figure 1).

The area under the receiver operating characteristic (ROC) curve of the BITE score was 0.839 (Figure 2). With an optimal cut-off point of 5, the BITE score was found to be ideal at predicting infections with an accuracy of 79.58%, a sensitivity of 82.31%, a specificity of 79.71%, a positive predictive value of 57.87%, and a negative predictive value of 92.38%. The Hosmer–Lemeshow statistic *p*-value was 0.016.

## 3. Discussion

Taiwan habus and green bamboo vipers account for two-thirds of venomous snakebites in Taiwan [1,2,3], though the literature regarding wound infections from these snakebites remains limited [8,9,10]. To the best of our knowledge, our study is the largest multicenter study to examine the bacteriology of snakebite wounds and determine the need for antibiotics.

We found that 22.45% of our snakebite patients had a wound infection, which agrees with the findings of 6–26% in previous reports [8,9,10,11]. None of these patients had a positive blood culture, suggesting that the snakebite infections remained localized without any systemic involvement or spread. The need for blood cultures is hence questionable in this patient population.

### 3.1. Association of Polymicrobial Infection, Snake Oral Cavity Bacteriology, and Venom Effect on Tissue

Up to 61 different pathogens have been identified in previous literature [11], of which, 12 were cultured in this study. All isolated pathogens, except *Stenotrophomonas maltophilia* and *Staphylococcus saprophyticus*, were skin commensals or opportunistic pathogens. A further review of the seven patients whose snakebite wounds were infected with opportunistic pathogens revealed that they were the most severely envenomed patients, requiring multiple surgical interventions throughout their hospital admission. On the contrary, other patients with cellulitis did not undergo surgical procedures, and as such, were not at risk of developing opportunistic infections. Regarding *Stenotrophomonas maltophilia* and *Staphylococcus saprophyticus*, the former frequently colonizes humid plastic surfaces, such as the tubes used in medical settings [12], while the latter is part of the normal flora in the female genital tract [13].

The infections of snakebites are often polymicrobial [8,9,10,11], with *Morganella morganii*, *Enterococcus* spp., *Bacteroides fragilis*, and *Aeromonas hydrophilia* being the most frequently isolated bacteria. Taking this into consideration and based on our antibiotic susceptibility data, a combination of ampicillin/oxacillin, fluoroquinolone, and clindamycin/metronidazole would be ideal for treating infected snakebites from the Taiwan habu and green bamboo viper.

Further analysis of the inpatient antibiotics regimen revealed that five out of the seven patients with positive swab cultures had their antibiotic therapy, and even class of antibiotics, partially or totally changed after the culture and sensitivity results were reviewed. This supports the value of our results in Table 2 and our suggested antibiotics choice for the empirical antibiotic treatment of snakebite patients with a high risk of developing wound infections.

The bacteriology of infected snakebite wounds in this study reflects the polymicrobial flora of snakes’ oral cavities. *Morganella morganii* and *Aeromonas hydrophilia* are frequently isolated from the oral cavity of snakes [14,15,16,17,18,19]. It is thus not surprising that while generally uncommon in soft tissue infections [9,10,11,14], they are two of the most common pathogens found in infected snakebite wounds. The transfer of bacteria from the snake’s oral cavity to the wound occurs via direct inoculation. During a snakebite from a Taiwan habu or a green bamboo viper, venom rich in metalloproteinases and phospholipase A2 is injected into the human tissue [20]. Phospholipase A2, which has cytotoxic, myotoxic, and pro-inflammatory properties [21], causes local tissue damage, allowing for the opportunistic colonization of the wound by snake oral cavity flora and skin commensals, as demonstrated in this study. The more venom that is injected, the more severe the resultant tissue damage and consequent infection. This might explain why patients with infected snakebite wounds in this study were administered more doses of antivenin compared to those without infected wounds.

### 3.2. Antibiotic Therapy in Infected Snakebite Wounds

Even though the positive swab cultures in our study were polymicrobial, a sizeable proportion of 44.8% of patients with infected snakebite wounds was only treated with single antimicrobial agents. The selection of the single-agent antibiotic therapy in this patient population was in accordance with our experience in empirically treating cellulitis with first-line agents of amoxicillin/clavulanic acid, oxacillin, cefazolin, and ampicillin/sulbactam. The fact that almost half of the patients did not require escalation to a poly-agent antibiotic therapy suggests that such treatment was adequate and effective. Based on our bacteriology and antibiotic susceptibility data, gentamicin and fluoroquinolones can be added to the first-line treatment in a stepwise manner, depending on the patient’s clinical condition, e.g., complicated wounds, septic shock, or multiple organ failure.

### 3.3. Factors Associated with a Secondary Bacterial Infection from Taiwan Habu and Green Bamboo Viper Snakebites

Although superimposed bacterial infections have been observed in ≈20% of snakebites from Taiwan habus and green bamboo vipers, there has yet to be any clinical predictive markers identified for this complication. Our BITE score was devised for this purpose.

The BITE score has two components: WBC × NLR and hospital admission. The first indicator can be found in the complete blood count/differential count, which is routinely used to check for infection. In our study, patients in the wound infection group had higher WBC counts (*p* = 0.0008) and NLRs (*p* = 0.01) than the non-wound-infection group. Leukocytosis indicates a higher probability for wound infection, possibly because of the patients’ inflammatory reaction to the primary snake venom insult and secondary infections persisting after antivenin therapy [22]. These physiological stressors also lead to raised neutrophil counts and decreased lymphocyte counts, resulting in elevated NLRs [23]. NLR has previously been shown to have high sensitivity and specificity within 6 h of inciting events, while WBC counts and band counts are less responsive [24,25]. As demonstrated in this study, WBC × NLR combines the utility of these two parameters into a single index that is more sensitive than either component alone.

More importantly, hospital admission was the most predictive factor for wound infection by far. The fact that infected snakebite patients who were hospitalized received more doses of antivenin than those who were treated as an outpatient suggests that the former group of patients were in a more clinically critical condition. Furthermore, wound infections were found to occur mostly in moderate or severe snakebite cases [22,26,27]. Therefore, it is reasonable to expect superimposed wound infections in patients with snakebites serious enough to warrant hospital admission.

### 3.4. Utility of the BITE Score

The BITE score is simple to use since it comprises only two parameters: hospital admission and the product of WBC × NLR. Both are easy to obtain clinically. There are two applications of our BITE score. Its primary use would be to enable the judicious use of antibiotics in snakebite patients based on local bacteriology data. The wound infection rate of only 23.3% in our patient population suggests that the common practice of routine prophylactic antibiotic administration in snakebite patients is unnecessary [7,28], representing a misuse of antibiotics. We thus propose that antibiotics only be given to those patients at high risk of developing wound infections (BITE score = 5).

The secondary use of the BITE score would be its employment in further studies to investigate the choice of antibiotic prophylaxis in snakebite patients. While prophylactic oral antibiotic administration is common in Taiwan, this practice remains controversial with no proper guidelines on the appropriateness of each antibiotic [28,29,30,31]. This proves problematic, as seen in our study, where more patients in the wound infection group received prophylactic antibiotics as opposed to those without wound infections, suggesting that the antibiotics chosen did little to prevent the onset of wound infection. Our findings complement those of previous studies that demonstrate the inefficacy of single-dose Augmentin, chloramphenicol, or benzylpenicillin with metronidazole at preventing secondary infections of snakebite wounds [22,32,33]. When taken into consideration vis-à-vis the fact that amoxicillin/clavulanic acid was one of the first-line antibiotics in our study for treating snakebite wounds with established infection, this suggests that the duration of the antibiotic regimen is perhaps as important as the antibiotic choice.

These findings can be extended to snakebite patients who were bitten by snakes other than a Taiwan habu or a green bamboo viper, as polymicrobial infections have been found in snakebite wounds of cobras, pit vipers, and *Bothrops* snakes, among others [17,33,34]. Poly-agent antibiotic prophylaxis might benefit the patient, even if the infection turns out to be monomicrobial, because it is not possible to determine the offensive pathogen in real time. This antibiotic prophylaxis should be avoided in snakebite patients with a low risk of wound infections.

The global snakebite burden is huge, with the highest burden of snakebite envenomation and death occurring in South Asia, Southeast Asia, and sub-Saharan Africa. For example, India reports 81,000 snake envenomation cases with nearly 11,000 deaths each year [35]. Our BITE score has substantial potential for improving treatment strategies for snakebite-associated infection in general. That said, the BITE score should be used with caution in patients bitten by other snakes before further studies are conducted due to the variations in clinical scenarios of envenomation by different species. Bites from the Taiwan habu and the green bamboo viper bites cause limb swelling without systemic effects on bleeding, venom-induced consumption coagulopathy, or acute kidney injury, as seen in victims of bites from other venomous snakes, such as Russell’s viper, *Trimeresurus albolabris*, or South Asian pit vipers. Nevertheless, our BITE score offers the medical community a framework for stratifying the risk of developing a wound infection after a snakebite, which can be improved upon and refined further. Future studies can be carried out to study the utility of antibiotic prophylaxis in snakebite patients at high risk of developing wound infections (BITE score = 5 in our patient group), as well as to determine the ideal combination of antibiotics to be used. The studied prophylactic antibiotic cocktail should be guided by local data on wound bacteriology and their antibiotic susceptibility.

### 3.5. Limitations

We have attempted to overcome the innate limitations of this retrospective study design (such as recall bias) by systematically retrieving data from the CGRD, which is based on original electronic medical records. Due to the anonymous nature of the CGRD, we were unable to differentiate between Taiwan habu or green bamboo viper snakebites in this study; however, this differentiation is unnecessary in the clinical context since both patient groups share the same management protocol. We have yet to validate the results of our BITE study, and as such, they require prospective validation in future trials. Nevertheless, the Hosmer–Lemeshow analysis of our BITE score supports the model’s stability.

Future studies may consider studying the profile of other circulatory inflammatory mediators to examine whether they might be a more sensitive and precise prognostic factor than the WBC × NLR indicator suggested by our study. If so, the incorporation of such an inflammatory mediator into the BITE score would further improve on its utility at predicting the development of wound infections in snakebite patients.

## 4. Conclusions

Wound infection was observed in 22.45% of the patients bitten by a Taiwan habus or a green bamboo viper. All positive swab cultures in this study were polymicrobial, of which *Morganella morganii*, *Enterococcus* spp., *Bacteroides fragilis*, and *Aeromonas hydrophila* were the most frequently cultured. Patients with a BITE score of 5 were at high risk of developing wound infections; we suggest that these patients be treated in a stepwise manner with the addition of gentamicin and fluoroquinolones to their antibiotic therapy, depending on clinical judgment of the severity of the wound infection. Our BITE score can also be used to identify patients for recruitment in future prospective clinical trials into the appropriate prophylactic antibiotic cocktail to be used when managing patients with snakebites.

## 5. Materials and Methods

### 5.1. Ethics Statement

This study was approved by the Chang Gung Memorial Hospital Research Ethical Committee, Taoyuan, Taiwan (Approval No: 201800736B0, date of approval: 21 May 2018), waiving the need for consent from study participants. All research methodologies were carried out in accordance with the relevant guidelines and regulations.

### 5.2. Data Resources and Setting

The variables analyzed in this study were retrieved from the computerized CGRD. The CGRD is a de-identified database derived from original medical records of the Chang Gung Memorial Hospital (CGMH) and systematically updated to include new data that is generated annually. Its overall coverage of 21.2% outpatient records and 12.4% inpatient records allows for conducting high-quality and scientifically sound studies [36].

The CGRD draws its strength from the large population CGMH serves. The hospital system, founded in 1976, is currently the largest in Taiwan, comprising seven medical institutes located in the northeastern to southern regions of Taiwan: Keelung CGMH, Taipei CGMH, Linkou CGMH, Taoyuan CGMH, Yunlin CGMH, Chiayi CGMH, and Kaohsiung CGMH. In total, the CGMH system has 10,070 beds with more than 280,000 hospital admissions each year, which accounts for 10.2% of hospitalized patients in Taiwan. Outpatient department visits and ED visits to CGMH numbered over 8,500,000 and 500,000 annually, respectively [37].

### 5.3. Enrolment of Patients

All victims of Taiwan habu or green bamboo viper snakebites who presented to the EDs of the CGMH and received at least one vial of FH antivenin (the designated antivenin for Taiwan habu and green bamboo viper snakebites) between January 2001 and January 2017 were identified via ICD-9 and ICD-10 codes for the diagnosis of a snakebite or the administration of antivenin. Patients receiving any other antivenin together with FH antivenin were excluded. Data were accessed anonymously and studied to develop our BITE score.

The variables collected include age, gender, and the results of the following laboratory investigations: complete blood count with differential counts (CBC/DC), coagulation profile, renal profile, liver profile, serum myoglobin, blood glucose, blood culture and sensitivity, and the culture and sensitivity of swabs taken from snakebite wounds. The doses of antivenin used, types of oral antibiotics, surgical procedures (debridement, fasciotomy, and graft), and length of hospitalization were also recorded. A polymicrobial infection was defined as the growth of two or more microbes on the same infected or purulent wound [38]. 

### 5.4. Management Protocol for Patients Presenting with Snakebites

All patients who present with a snakebite to our EDs are managed in accordance with World Health Organization guidelines [39]. The choice of antivenin, if indicated, is dependent on the species of venomous snake; therefore, patients will be asked to identify it through a pictorial chart available in our EDs.

Given the lack of specific international guidelines regarding the admission criteria for snakebite patients after the initial treatment in the ED, our ED panel of experts have decided on the following management protocol based on local experience in treating snakebites: patients are monitored in the ED for 24 h post-bite till there is a clinical improvement in limb pain and swelling, after which, those who are hemodynamically stable with no coagulopathy are allowed to go home. The remaining patients will be admitted to our wards for further observation and treatment. If limb swelling reappears after adequate antivenin administration, or if there are clinical signs of cellulitis, these patients will be diagnosed with a wound infection and likewise admitted to our wards. This protocol remained consistent throughout our study period.

### 5.5. Definition of Wound Infection

Patients were deemed to have infected snakebite wounds if they fulfilled one of the following criteria: (1) had positive wound cultures, (2) admission diagnoses of cellulitis or an abscess, or (3) underwent surgical wound debridement. Patients who received oral antibiotic prophylaxis on the day of the snakebite, as is commonly practiced in Taiwan [28], without meeting these criteria were not considered to have had wound infections.

### 5.6. Statistical Analysis and Development of the BITE Score

Categorical variables are reported as frequencies and percentages, while continuous variables are expressed as means ± SEM, unless indicated otherwise. Univariate analyses were conducted using Student’s *t*-tests for numerical variables and chi-squared tests for categorical variables, with odds ratios calculated to assess the strength of the association. Variables with *p* < 0.1 were identified for further multivariate analysis using multiple logistic regression. Our BITE score was then constituted by weighting these variables according to their β coefficients.

The BITE scores for each patient were subsequently calculated and used to plot a ROC curve. The area under the curve (AUC) analysis was done to examine the accuracy of the BITE score at predicting the need for antibiotics, with the optimal cut-off point identified using Youden’s index. A *p*-value < 0.05 was regarded as statistically significant. All statistical analyses were performed with SAS statistical software version 9.2 (SAS Institute, Cary, NC, USA, 2013).

## Figures and Tables

**Figure 1 toxins-12-00575-f001:**
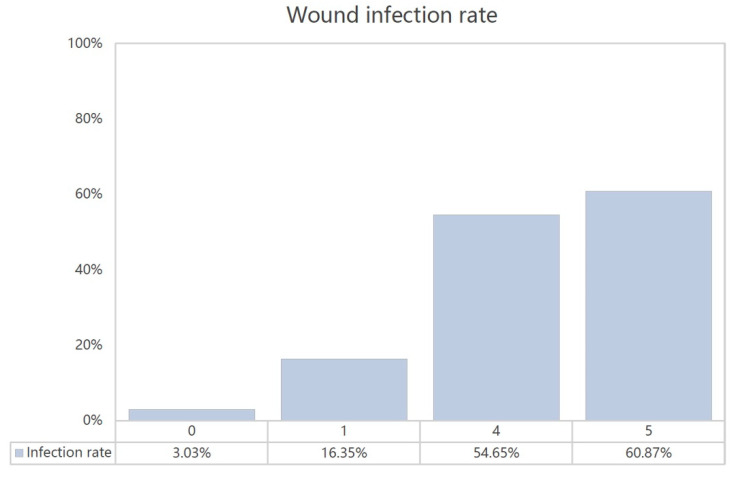
The BITE scores and wound infection rates.

**Figure 2 toxins-12-00575-f002:**
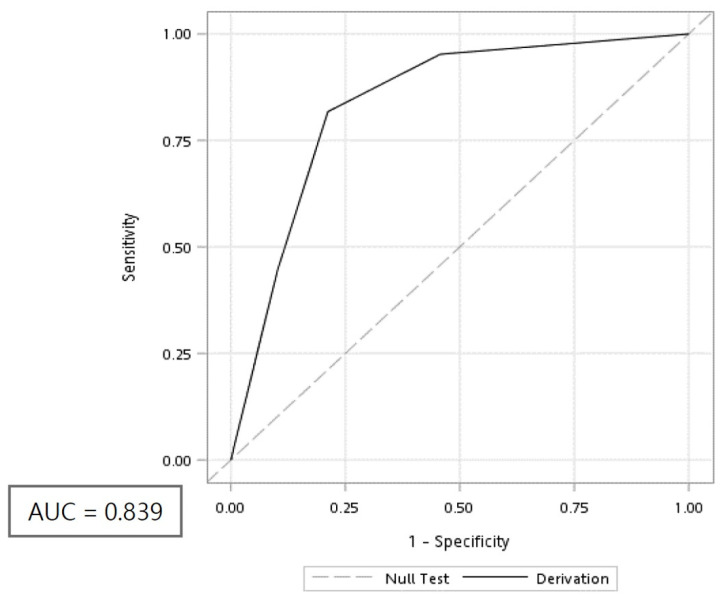
Receiver operating characteristic (ROC) curve of the BITE score. AUC: Area under the curve.

**Table 1 toxins-12-00575-t001:** Univariate analysis of patient demographics, laboratory results, and treatment modalities.

Variable	Patients		*p*-Value
	No Wound Infection (*n* = 563)	Wound Infection (*n* = 163)	
Demographic Characteristics			
Age	51.88 (17.42)	52.76 (18.04)	0.57
Male, *n* (%)	400 (71.05)	106 (65.03)	0.17
Laboratory Variables ^#^			
WBC (1000/μL) *	8.15 (3.59)	9.47 (3.97)	0.0008
Band	0.07 (0.72)	0.11 (0.74)	0.58
Segment *	62.26 (14.03)	67.54 (14.33)	0.0003
Lymphocyte *	29.59 (12.43)	24.63 (12.53)	0.0001
Neutrophil/lymphocyte ratio *	3.26 (3.63)	4.56 (5.19)	0.01
Hb (g/dL)	14.35 (1.66)	14.08 (1.74)	0.11
RDW	18.38 (11.18)	18.98 (11.70)	0.60
PLT (1000/μL)	211.60 (61.00)	214.50 (62.75)	0.63
Prothrombin time	11.69 (7.11)	11.37 (3.15)	0.49
APTT	28.16 (7.14)	27.38 (3.59)	0.11
Cr (mg/dL)	0.93 (0.57)	0.94 (0.97)	0.90
BUN (mg/dL)	14.98 (8.20)	14.72 (6.99)	0.80
ALT/GPT (U/L)	27.44 (29.31)	23.42 (12.87)	0.06
AST/GOT	33.17 (36.37)	40.49 (59.34)	0.39
Creatine kinase	388.00 (520.50)	1201.40 (3895.60)	0.24
Myoglobin	168.90 (417.30)	245.20 (770.40)	0.49
K (mEq/L)	3.74 (0.38)	3.78 (0.43)	0.45
Na (mEq/L) *	140.00 (2.11)	139.10 (2.51)	0.0022
Glucose (mg/dL)	130.60 (47.57)	144.00 (53.49)	0.11
Treatment Modalities			
Antivenin dose (vial) ^&^*	2 (1–3)	2 (1–4)	<0.0001
Prophylactic antibiotics *	215 (38.6)	106 (62.72)	<0.0001
Hospitalization			
Hospital admission, *n* (%) *	98 (17.41)	132 (80.98)	<0.0001
Length of hospitalization (day) *	4.87 (3.50)	9.50 (8.04)	<0.0001

WBC: white blood cell; Hb: hemoglobin; RDW: red blood cell distribution width; PLT: platelet; APTT: activated partial prothrombin time; BUN: blood urea nitrogen; ALT/GPT: alanine aminotransferase/glutamic pyruvic transaminase; AST/GOT: aspartate aminotransferase/glutamic oxaloacetic transaminase; ^#^ Continuous variables expressed as mean (SEM); ^&^ antivenin dose is expressed in vials as median (IQR); * indicates statistical significance.

**Table 2 toxins-12-00575-t002:** Wound bacterial culture and sensitivity results.

No.	Microorganism			Suggested Antibiotic Sensitivity	
	Aerobic Gram-Positive	Aerobic Gram-Negative	Anaerobic		
1	*Enterococcus faecalis*	*Morganella morganii*	*Bacteroides fragilis*	Ampicillin	Clindamycin
2	*Enterococcus faecalis*	*Morganella morganii*		Ampicillin	
		*Aeromonas hydrophila*		Ceftriaxone	
3	*Enterococcus faecalis*	*Morganella morganii*		Ampicillin	
4	*Corynebacterium jeikeium*		*Enterobacter cloacae* CR strain	Fluoroquinolone	
5	*Coagulase negative* *Staphylococcus*			Vancomycin	
6	*Viridans streptococcus*			Oxacillin	
	*Staphylococcus aureus*			Vancomycin	
	*Staphylococcus saprophyticus*	*Stenotrophomonas maltophilia*	*Enterobacter cloacae*	Fluoroquinolone	
			*Peptostreptococcus micros*		
7	*Staphylococcus aureus*			Oxacillin	
	*Corynebacterium jeikeium*			Fluoroquinolone	
	*Corynebacterium* spp.				

CR: carbapenem-resistant.

**Table 3 toxins-12-00575-t003:** Multiple logistic regression and the Bacteriology of Infections in Taiwanese snake Envenomation (BITE) score.

Variable	β	Odds Ratio	95% Confidence Interval	Points ^※^
Intercept	−1.06			
WBC (1000/μL) × neutrophil-lymphocyte ratio ≥ 19.84	0.37	2.112	(1.289, 3.462)	1
Admission	1.38	15.65	(9.27, 26.42)	4

^※^ 0.4β = 1 point.

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
