# Peer review of "Wound Infections of Snakebites from the Venomous Protobothrops mucrosquamatus and Viridovipera stejnegeri in Taiwan: Bacteriology, Antibiotic Susceptibility, and Predicting the Need for Antibiotics—A BITE Study"

_toxins, 2020, doi:10.3390/toxins12090575_

Round 1

Reviewer 1 Report

Reviewer:

General Comments:

Thank you for the opportunity to review this article about wound infections of snakebites from the venomous Protobothrops mucrosquamatus and Viridovipera  stejnegeri in Taiwan: bacteriology, antibiotic susceptibility, and predicting the need for antibiotics  – BITE study.

This article focuses on the treatment of wound infections following  2 venomous snakebites in Taiwan. This is a very interesting national snakebite protocol, probably very important for the management of bacterial wound infections following snakebites in Taiwan, in which snake bite is a common and very important health concern.

This article described a  7-year descriptive analysis of snake bites in Taiwan hy 2 venomous snakes, and evaluation of the protocolled administration of Anti Snake Venom several hospitals. Out of 726 patients presented with snakebites from the Taiwan habu and green bamboo viper. The wound infection rate was 22.45%, with 7 positive wound cultures including  6 polymicrobial infections. Morganella morganii, Enterococcus species, Bacteroides fragilis, and Aeromonas hydrophila were most frequently cultured.

The author pointed out several findings and few limitations. This paper also reported some useful information about the management and the clinical features of snakebites by using the score BITE.

I recommend minor revision. I would like to suggest some corrections that could be useful for the revision.

Concerns:

  1. Can you describe the clinical criteria to define infection in the 7 positive wound infection patients, the grade of envenomation, and the severity of such cases?
  2. Since there were no positive blood cultures. 67.0% of snakebite patients who received prophylactic antibiotics nevertheless developed wound infections. Wound infection following snakebite usually accounts for 10 to 77% of the bitten patients, as described in several other studies. I believe that a high proportion of microbiological cultures were negative in your study because of systematic preemptive use of antibiotics in snake-bitten patients. Are you agree?

  1. Why these patients had to be treated with prophylactic antibiotics and why did you use amoxicillin/clavulanic acid instead of the cephalosporine third generation, since Aeromonas hydrophilia is naturally resistant to amoxicillin/clavulanic?

  1. Based on our bacteriology and antibiotic susceptibility data, gentamicin and fluoroquinolones can be added on to the first-line treatment in a stepwise manner depending on the patient’s clinical condition. So, in which group of patients should we use bi-therapy? In the case of septic shock or circulatory failure or multiple organ failure?

This review describes significant results and deserves publication after minor revisions.

Author Response

Dear Reviewer #1,

Thank you for your precious suggestions to our manuscript. Below are our responses to your comments. All amendments have been marked in RED.

  1. Can you describe the clinical criteria to define infection in the 7 positive wound infection patients, the grade of envenomation, and the severity of such cases?

 The following has been added [Lines 80-81]: “The patients with the most severe envenomation consequently had the most severe infected wounds which required surgical debridement.”

  1. Since there were no positive blood cultures. 67.0% of snakebite patients who received prophylactic antibiotics nevertheless developed wound infections. Wound infection following snakebite usually accounts for 10 to 77% of the bitten patients, as described in several other studies. I believe that a high proportion of microbiological cultures were negative in your study because of systematic preemptive use of antibiotics in snake-bitten patients. Are you agree?

We agree with the reviewer’s analysis, and have added the following [Lines 115-117]: “The yield of positive microbiological cultures in our study is much lower than expected, especially when compared with other studies. This could perhaps be attributed to the systematic pre-emptive use of antibiotics in snakebite patients on ED presentation in our setting.”

  1. Why these patients had to be treated with prophylactic antibiotics and why did you use amoxicillin/clavulanic acid instead of the cephalosporine third generation, since Aeromonas hydrophilia is naturally resistant to amoxicillin/clavulanic? 

Thank you for pointing this out. We have changed our suggested antibiotic for Aeromonas hydrophila to ceftriaxone. [Table 2]

  1. Based on our bacteriology and antibiotic susceptibility data, gentamicin and fluoroquinolones can be added on to the first-line treatment in a stepwise manner depending on the patient’s clinical condition. So, in which group of patients should we use bi-therapy? In the case of septic shock or circulatory failure or multiple organ failure?

We have added on the following [Lines 217-219]: “Based on our bacteriology and antibiotic susceptibility data, gentamicin and fluoroquinolones can be added on to the first-line treatment in a stepwise manner depending on the patient’s clinical condition, e.g. complicated wounds, septic shock, or multiple organ failure.

Reviewer 2 Report

This is a study who aims to investigate the prevalence of wound infection, bacteriology and antibiotic usage in patients who was snakebited from the gender Protobothrops mucrosquamatus and Viridovipera stejnegeri in Taiwan. Despite being an important study there is a lack of information in the introduction section and in tables and detailed results descriptions.

Several major points must be addressed:

  1. The introduction section has is a huge lack of information to support the reading of the manuscript.
  2. It is not clear if authors aim to propose a treatment or prophylaxis with antibiotics
  3. The number of 7 patients, showed in table 1 is small to support the article idea.
  4. The authors do not even mention the side effects and risks of prophylactic use of antibiotics.
  5. The tables are not clear.
  6. It is not clear how authors stratified the wound infection rate and developed the BITE score. Taking only WBC counts is a poor method to measure inflammation. Authors should include in the analysis the profile of circulating inflammatory mediators.
  7. In the discussion section authors point out that the practice of prophylactic antibiotics is unnecessary, however this is one of the main aims raised by the study.

Author Response

Dear Reviewer #2,

Thank you for your precious suggestions to our manuscript. Below are our responses to your comments. All amendments have been marked in RED.

We have also asked a native English speaker to review our manuscript as requested. In the event you are of the opinion that it requires further editing, kindly let us know and please grant us a deadline extension so that we may send it to a language editing service as you have suggested.

  1. The introduction section has is a huge lack of information to support the reading of the manuscript.

We have added the following to further characterize the need for a prognostic scoring system in the management of snakebites.

[Lines 50-61]: “Since clinical differentiation between envenomation and wound infection is difficult especially in early presentations, such snakebites are often treated empirically with prophylactic antibiotics [5] in addition to the freeze-dried hemorrhagic (FH) antivenin [6-7]. This practice of administering prophylactic antibiotics is widespread, in line with the general management of animal bites to prevent the development of subsequent wound infections.

Up to 80% of these snakebite patients are not eventually diagnosed with cellulitis or wound infection [5], representing a gross overuse of antibiotics in this patient population. This issue arises from the current lack of evidence-based guidelines or tools to guide proper antibiotic therapy in snakebite patients. As such, developing a suitable tool to predict probability of snakebite wound infection is crucial in augmenting the management of snakebite complications. Said prognostic tool may also be employed in further studies to investigate the choice of antibiotic prophylaxis in snakebite patients.

  1. It is not clear if authors aim to propose a treatment or prophylaxis with antibiotics 

As stated in our introduction, we embarked on this study to investigate prevalence of snakebite wound infections, bacteriology, and antibiotic susceptibility profile, with the ultimate goal of developing a prognostic scoring system to help evaluate the risk of developing infections. The higher the risk, the stronger the indication for antibiotic prophylaxis. We aim to provide a framework for clinicians to make this decision, so as to avoid administering antibiotics unnecessarily.

The following was added [Lines 64-67]: “The ultimate aim was to develop a prognostic evaluation tool to guide antibiotic usage in patients with snakebites from the Taiwan habu and the green bamboo viper – the higher the risk of infection, the stronger the indication for antibiotic prophylaxis. This would then provide clinicians with a framework to prescribe antibiotics judiciously.”

  1. The number of 7 patients, showed in table 1 is small to support the article idea.

We respectfully disagree with the reviewer on this point. Our statistical analysis is based on analysis of 726 patients. The 7 patients referred to by the reviewer are those with positive cultures – we suggested antibiotics based on their positive isolates. This is however not the main thrust of our manuscript. We would like to reiterate that our main aim is to provide a scoring system to evaluate risk of developing infections in snakebites, and hence allow clinicians to judiciously prescribe antibiotics. This would help to cut down on the practice of routinely and possibly unnecessarily administering prophylactic antibiotics in snakebite patients.

  1. The authors do not even mention the side effects and risks of prophylactic use of antibiotics. 

We would like to clarify that we are not proposing a completely new practice. As of current clinical practice in our setting, prophylactic antibiotic use is routinely administered in all snakebite patients regardless of risk of developing wound infections. Our manuscript deals with evaluating this risk to help clinicians decide if antibiotic prophylaxis is really necessary. We have therefore not discussed the side effects and risks of antibiotic prophylaxis as we are advocating judicious (and possibly a reduction in) antibiotic prescription, rather than the introduction of a brand-new antibiotic prophylaxis regimen. The follow paragraphs were added to clarify:

[Lines 52-54]: “This practice of administering prophylactic antibiotics is widespread, in line with the general management of animal bites to prevent the development of subsequent wound infections.

[Lines 64-67]: “The ultimate aim was to develop a prognostic evaluation tool to guide antibiotic usage in patients with snakebites from the Taiwan habu and the green bamboo viper – the higher the risk of infection, the stronger the indication for antibiotic prophylaxis. This would then provide clinicians with a framework to prescribe antibiotics judiciously.

  1. The tables are not clear.

Table 1 provides the results of univariate analysis of patient demographics, laboratory results, and treatment modalities in patients without infected snakebite wounds versus patients with infected snakebite wounds. Table 2 lists out all microorganisms isolated from swab cultures in our patient population, together with our suggested antibiotics for each microorganism. Table 3 tabulates the results of multiple logistic regression of each parameter with respect to developing infection. We hope this clarifies any confusion the reviewer has.

  1. It is not clear how authors stratified the wound infection rate and developed the BITE score. Taking only WBC counts is a poor method to measure inflammation. Authors should include in the analysis the profile of circulating inflammatory mediators.

We would like to point the reviewer’s attention to the section 5.6 in our manuscript:

“Categorical variables were reported as frequencies and percentages, while continuous variables were expressed as means ± SEM unless indicated otherwise. Univariate analyses were conducted via student’s t-test for numerical variables and chi-squared test for categorical variables, with odds ratios calculated to assess the strength of association. Variables with p<0.1 were identified for further multivariate analysis using multiple logistic regression. Our BITE score was then constituted by weighting these variables according to their β coefficients. BITE scores for each patient were subsequently calculated and used to plot a ROC curve. Area under the curve (AUC) analysis was done to examine the accuracy of BITE in predicting need for antibiotics, with optimal cut-off point identified via Youden’s index. A p-value of <0.05 was regarded as statistically significant. All statistical analyses were performed with SAS statistical software version 9.2 (SAS Institute, Cary, NC, USA).”

We agree that WBC alone is an unsatisfactory indicator of inflammation. Hence we have combined it with NLR to increase the sensitivity of our BITE score (section 3.3). Thank you for your suggestion of including analyses of circulating inflammatory mediators, we have added the following paragraph:

[Lines 291-295]: “Future studies may consider studying the profile of other circulatory inflammatory mediators to examine if they might turn out to be a more sensitive and precise prognostic factor than the WBC×NLR indicator suggested by our study. If so, the incorporation of such an inflammatory mediator into BITE score would further improve on its utility in predicting the development of wound infections in snakebite patients.

  1. In the discussion section authors point out that the practice of prophylactic antibiotics is unnecessary, however this is one of the main aims raised by the study.

We would like to clarify that we mean the routine use of prophylactic antibiotics is perhaps unnecessary, hence the need for a prediction rule like our BITE score. We have added the following to clarify [Lines 245-247]: “The wound infection rate of only 23.3% in our patient population suggests that the common practice of routine prophylactic antibiotic administration in snakebite patients is unnecessary [7,28], representing a misuse of antibiotics.”

Reviewer 3 Report

The article entitled "Wound infections of snakebites from the venomous Protobothrops mucrosquamatus and Viridovipera stejnegeri in Taiwan: bacteriology, antibiotic susceptibility, and predicting the need for antibiotics – BITE study" presents a very interesting and important approach designed to guide the treatment of snake bite victims. The article is very-well written, and the only minor comments regarding the data should be better addressed.

I suggest that the term BITE could be better explained in the abstract.

One point that is really important is the table 1 where is presented the bacteria associated with snake bite. I understand that the data where accessed from hospital data banks, but it is intriguing that the antibiotics suggested were those with a large specter. Did you get any information if the antibiotics regimen was changed after the bacteriology results?

In table 2, looking at the data, it is hard to understand how low is the p value considering the SD of the samples, for example NLR no wound 3.26(3.63) vs wound 4.56(5.19). Have the authors considered using SEM (standard error of mean) instead? 

On topic 5.4. it is mentioned the WHO guidelines, it would be important to include it in the references.

Author Response

Dear Reviewer #3,

Thank you for your critical review and valuable suggestions. Below are our responses to your comments. All amendments have been marked in RED.

  1. The article entitled "Wound infections of snakebites from the venomous Protobothrops mucrosquamatus and Viridovipera stejnegeri in Taiwan: bacteriology, antibiotic susceptibility, and predicting the need for antibiotics – BITE study" presents a very interesting and important approach designed to guide the treatment of snake bite victims. The article is very-well written, and the only minor comments regarding the data should be better addressed. I suggest that the term BITE could be better explained in the abstract.

Thank you for your comments. The term BITE is a backronym coined by our authors with hopes that it would be an easy-to-remember name for our proposed scoring tool. It does not specifically have any clinical significance, except perhaps to record the origins of this proposed score if or when it becomes widely used in the management of all snakebites

  1. One point that is really important is the table 1 where is presented the bacteria associated with snake bite. I understand that the data where accessed from hospital data banks, but it is intriguing that the antibiotics suggested were those with a large specter. Did you get any information if the antibiotics regimen was changed after the bacteriology results?

We have gone through our data again to review any changes in antibiotics regimen, and found that most of them required such changes. Thank you for this invaluable perspective that further supports the value of our results. We have added the following paragraph:

[Lines 193-197]: “Further analysis of inpatient antibiotics regimen revealed that 5 out of 7 patients with positive swab cultures had their antibiotic therapy, and even class of antibiotics, partially or totally changed after culture and sensitivity results were reviewed. This supports the value of our results in Table 2 and our suggested antibiotics choice for empirical antibiotic treatment of snakebite patients with high risk of developing wound infections.

  1. In table 2, looking at the data, it is hard to understand how low is the p value considering the SD of the samples, for example NLR no wound 3.26(3.63) vs wound 4.56(5.19). Have the authors considered using SEM (standard error of mean) instead? 

After careful re-examination of our statistical data, we found that we have mistakenly used SD instead of SEM in our manuscript. We want to apologize this mistake and thank the reviewer catching this error. We have corrected this mistake in the revised version. 

  1. On topic 5.4. it is mentioned the WHO guidelines, it would be important to include it in the references.

We have added the WHO guidelines as reference #39.

Round 2

Reviewer 2 Report

The auhors addressed the issues raised. 

Author Response

Dear Reviewer,

Thank you for your comments that we have addressed all the issues raised.